# S-72, a Novel Orally Available Tubulin Inhibitor, Overcomes Paclitaxel Resistance via Inactivation of the STING Pathway in Breast Cancer

**DOI:** 10.3390/ph16050749

**Published:** 2023-05-15

**Authors:** Zhenyan Hou, Songwen Lin, Tingting Du, Mingjin Wang, Weida Wang, Shen You, Nina Xue, Yichen Liu, Ming Ji, Heng Xu, Xiaoguang Chen

**Affiliations:** 1State Key Laboratory of Bioactive Substance and Function of Natural Medicines, Institute of Materia Medica, Chinese Academy of Medical Sciences and Peking Union Medical College, Beijing 100050, China; houzhenyan@imm.ac.cn (Z.H.); linsongwen@imm.ac.cn (S.L.); ninadu@imm.ac.cn (T.D.); wangmingjin@imm.ac.cn (M.W.); wangweida@imm.ac.cn (W.W.); youshen@imm.ac.cn (S.Y.); angelnina@imm.ac.cn (N.X.); 2Beijing Key Laboratory of New Drug Mechanisms and Pharmacological Evaluation Study, Institute of Materia Medica, Chinese Academy of Medical Sciences and Peking Union Medical College, Beijing 100050, China; 3Beijing Key Laboratory of Non-Clinical Drug Metabolism and PK/PD Study, Institute of Materia Medica, Chinese Academy of Medical Sciences and Peking Union Medical College, Beijing 100050, China

**Keywords:** microtubule-destabilizing agent, breast cancer, paclitaxel resistance, STING, chromosomal instability

## Abstract

Microtubule-targeting agents are widely used as active anticancer drugs. However, drug resistance always emerges after their long-term use, especially in the case of paclitaxel, which is the cornerstone of all subtypes of breast cancer treatment. Hence, the development of novel agents to overcome this resistance is vital. This study reports on a novel, potent, and orally bioavailable tubulin inhibitor called S-72 and evaluated its preclinical efficacy in combating paclitaxel resistance in breast cancer and the molecular mechanisms behind it. We found that S-72 suppresses the proliferation, invasion and migration of paclitaxel-resistant breast cancer cells in vitro and displays desirable antitumor activities against xenografts in vivo. As a characterized tubulin inhibitor, S-72 typically inhibits tubulin polymerization and further triggers mitosis-phase cell cycle arrest and cell apoptosis, in addition to suppressing STAT3 signaling. Further studies showed that STING signaling is involved in paclitaxel resistance, and S-72 blocks STING activation in paclitaxel-resistant breast cancer cells. This effect further restores multipolar spindle formation and causes deadly chromosomal instability in cells. Our study offers a promising novel microtubule-destabilizing agent for paclitaxel-resistant breast cancer treatment as well as a potential strategy that can be used to improve paclitaxel sensitivity.

## 1. Introduction

According to the cancer statistics from GLOBOCAN 2020, breast cancer has exceeded lung cancer as the most commonly diagnosed cancer and ranks as the leading cause of cancer death among women in 110 countries [1]. While new targeted therapies and immunotherapies are revolutionizing breast cancer treatments, chemotherapy still remains the major option for breast tumors. Microtubule-targeting agents (MTAs) are among the most active drugs for breast cancer treatment, among which paclitaxel is the fundamental drug, being widely used in all subtypes of breast tumors [2]. However, therapeutic resistance often occurs and dampens the efficacy of chemotherapy. Hence, the development of novel agents or strategies to overcome paclitaxel resistance is vital.

With regard to the mechanisms of paclitaxel resistance in breast cancer, multiple factors are reported to be involved, such as the overexpression of the multidrug transporter P-glycoprotein (P-gp), altered cellular expression of the paclitaxel binding site, β-tubulin, especially βIII-tubulin, and expressional changes in the microtubule-associated protein TAU, which binds to the same site occupied by paclitaxel in microtubules [3,4,5]. The overexpression of oncogene products also contributes to paclitaxel resistance, among which the transcription factor signal transducer and activator of transcription 3 (STAT3) is frequently overexpressed and activated in paclitaxel-resistant breast cancer and is associated with tubulin [6].

A recent study showed that chromosomal instability (CIN) is positively correlated with paclitaxel sensitivity [7]. Since paclitaxel targets cell microtubules, the following multipolar divisions may cause ongoing chromosome mis-segregation, which indicates CIN. Moderate CIN leads to alternative oncogenic driver generation, thus endowing cancer cells with the ability to evolve and resist chemotherapy, while excessive CIN may provide cancer cells with unique vulnerabilities, thus overcoming drug resistance [8,9]. Based on this observation, the eradication of this crucial dependency for survival in resistant cells with CIN could break the balance and improve sensitivity. The stimulator of interferon genes (STING), a crucial transducer of type I interferon (IFN) production triggered by the cytoplasmic accumulation of double-stranded DNA (dsDNA), was reported to maintain balance and cell survival via the regulation of CIN [10,11].

Unlike paclitaxel, colchicine binding site inhibitors (CBSIs) are at less risk of therapeutic resistance. Currently, several novel colchicine binding site inhibitors (CBSIs) are under exploration in studies aiming to combat paclitaxel resistance due to the fact that their binding sites in tubulin are different from that of paclitaxel, as well as their general characteristics and the fact that they are not transported by P-gp [12,13,14,15,16]. However, to date, none of these CBSIs have gained FDA approval, mainly due to their toxicity and lack of clinical benefits, problems that are related to their specific chemical structures. Compounds such as CA-4P, VERU-111, OXi4503, ABT-751, BNC105P and ZD6126 are currently undergoing or have completed clinical trials [17,18,19,20], but none are currently approved for clinical applications in cancer treatment due to their causation of undesirable adverse events [21], lack of bioavailability and low aqueous solubility [22,23]. Some other compounds are in preclinical studies. Most are CA-4 or VERU-111 analogues with known parent nucleus structures or novel structural compounds such as SB226, which is administered via intraperitoneal or intravenous injection [16,24,25]. The development of a new generation of CBSIs with unique chemical structures is urgently required for safe and effective chemotherapy, particularly in the case of paclitaxel-resistant breast cancers. In the present study, we identified a novel orally available tubulin inhibitor, S-72, which combats paclitaxel resistance in breast cancer both in vitro and in vivo. We intended to elucidate the molecular mechanisms in vitro, thus providing a basis for S-72’s use as a promising novel microtubule-destabilizing agent for paclitaxel-resistant breast cancer treatment, as well as a potential strategy that can be used to improve paclitaxel sensitivity.

## 2. Results

### 2.1. S-72 Exhibited Potent Cytotoxicity in Paclitaxel-Resistant Breast Cancer Cells

The cytotoxicity of S-72 was tested in the paclitaxel-resistant human breast cancer cell lines MCF7/T and MX-1/T and their parents using CCK8 assays. S-72, as well as paclitaxel and colchicine, showed excellent inhibitory activity in the nanomolar range against the non-resistant parental breast cells (Figure 1B). In paclitaxel-resistant MCF7/T and MX-1/T, S-72 still presented favorable cytotoxicity at nanomolar concentrations, with resistance indices (RIs) of only 1.68 and 9.11, respectively, while both paclitaxel and colchicine were tolerated at high RI values. Cell morphology analysis also validated the hypothesis that S-72 displays high sensitivity in both MCF7 and MCF7/T cells (Figure 1C). Moreover, S-72 dose-dependently repressed colony formation in both the MCF7 and MCF7/T cells (Figure 1D). Paclitaxel and colchicine robustly inhibited colony formation in the parental cells, but their effects on the resistant cells were weaker.

### 2.2. S-72 Suppressed Both the Invasion and Migration of MCF7 and MCF7/T Cells

In addition to the anti-proliferative effect, cell invasion and migration were also evaluated after the cells’ exposure to S-72. As shown in Figure 2A, S-72 inhibited both the invasion and migration of MCF7 and MCF7/T cells in a concentration-dependent manner. A wound healing assay also verified that compared with the control group, S-72, as well as colchicine, significantly inhibited the migration of both the MCF7 and MCF7/T cells, with less wound closure, while paclitaxel showed lower efficacy, especially in the MCF7/T cells (Figure 2B).

### 2.3. S-72 Inhibited Tubulin Polymerization, Triggered Mitosis-Phase Cell Cycle Arrest and Cell Apoptosis and Disrupted STAT3 Signaling Activation

The inhibitory effect of S-72 on microtubule assembly dynamics was explored via the in vitro tubulin polymerization assay. As shown in Figure 3A, S-72 inhibited the rate and extent of microtubule polymerization in a concentration-dependent manner. In our comparison, colchicine suppressed tubulin polymerization, whereas paclitaxel enhanced tubulin polymerization, and the tubulin polymerization activity of S-72 was superior to that of colchicine at the same concentrations. As a tubulin polymerization inhibitor, the effects of S-72 on microtubule stabilities were further validated by determining the protein level of acylated-α-tubulin (Lys40), which has been reported to stabilize microtubules. The results showed that S-72 decreased the levels of acylated-α-tubulin in both the parental and paclitaxel-resistant breast cancer cells (Figure 3B), indicating that S-72 interfered with the microtubule stabilities.

Since MTAs generally induce G2/M cell cycle arrest and further apoptosis, the effects of S-72 on cell cycle disruption and cell apoptosis were evaluated in both the parental and paclitaxel-resistant cells. As displayed in Figure 3C–E, S-72 significantly blocked the G2/M phase in both the MCF7 and MCF7/T cells, as did colchicine. Paclitaxel induced G2/M arrest in the MCF7 cells but not the MCF7/T cells. In addition, the proportion of polyploid (>4N DNA) cells increased in the pair of cells, as well as the paclitaxel-treated MCF7 cells, after S-72 or colchicine treatment. Correspondingly, the key molecular events during S-72-induced cycle arrest were evaluated via Western blot. The results showed that S-72 induced the upregulation of p-cdc25c (Thr48) and cdc25c and the downregulation of p-cdc2 (Tyr15) in the MCF7 and MCF7/T cells, and the effect was stronger than that of colchicine. It upregulated cdc2, cyclin B1 and p-histone H3 (Ser10) and ultimately led to mitosis phase arrest (Figure 3F).

Next, the effects of S-72 on cell apoptosis were evaluated via flow cytometry and Western blot. S-72 increased the proportion of apoptotic cells in both the MCF7 and MCF7/T cells, and its effect on the MCF7/T cells was stronger than that of colchicine, while paclitaxel induced apoptosis in the MCF7 cells but not the MCF7/T cells (Figure 3G–I). Accordingly, S-72 upregulated the expression of cleaved caspase-3 and cleaved PARP and downregulated the expression of bcl-2 (Figure 3J).

Furthermore, STAT3 signaling was detected due to the typical disruption of STAT3/tubulin interactions and further STAT3 function blocking by MTAs. S-72 dramatically reduced the phosphorylation levels of p-STAT3 (Tyr705) and increased the p-STAT3 (Ser727) levels, and the variations were more significant than those observed in the colchicine-treated group (Figure 3K). S-72 also inhibited IL-6-induced STAT3 activation at Tyr705. Furthermore, S-72 showed no effects on STAT3 upstream JAK2 activation (Appendix A).

### 2.4. S-72 Repressed the Growth of Both Paclitaxel-Sensitive and Paclitaxel-Resistant Human Tumor Xenografts In Vivo

As S-72 exhibited potent cytotoxicity in vitro, we further evaluated its antitumor activity in vivo. In an MCF7/T mouse xenograft model, tumor growth was significantly inhibited by S-72 in a dose-dependent manner. The TGI of the S-72 group was 60.1% at the dose of 10 mg/kg. No significant difference in the tumor size or weight between the paclitaxel and vehicle groups was observed (Figure 4A,B). Similar trends were observed in the MX-1/T xenograft model, with the TGI of 87.8% at 10 mg/kg in the S-72 group (Figure 4D,E). H&E staining showed that compared with the vehicle group, a larger necrosis area and less cell proliferation were observed in the S-72 treatment groups, and these effects were further confirmed using Ki-67 antibody staining. Paclitaxel inhibited the rate of Ki-67-positive cells in the tumors of the parental—but not the resistant—cell-bearing mice (Figure 4C,F). Excellent antitumor activities of S-72 were also shown in the parental mouse xenograft models (Figure 4G–L).

### 2.5. S-72 Inhibited STING Activation in Paclitaxel-Resistant Cancer Cells

In consideration of the excellent antitumor activity of S-72, we aimed to identify the potential mechanism employed by S-72 to overcome paclitaxel resistance. P-gp, βIII-tubulin and TAU have been reported to be involved in paclitaxel resistance. Our previous study also demonstrated that P-gp is highly expressed in both MCF7/T and MX-1/T cells [26]. Thus, we investigated whether P-gp is affected by S-72. The results showed that S-72 had no influence on P-gp in terms of the protein levels (Figure 5A). Moreover, compared with the MCF7 cells, the MCF7/T cells did not express excess βIII-tubulin or TAU, and likewise, S-72 did not change the expressions of these two proteins (Figure 5B). According to the flow cytometry analysis, much lower levels of Rh-123, a classical P-gp substrate, were accumulated in the MCF7/T and MX-1 cells compared with their respective parental cells (Figure 5C). Verapamil, a known P-gp inhibitor, significantly enhanced the accumulation of Rh-123 in both the MCF7 and MCF7/T cells, while S-72 showed inactive effects; thus, it was not considered as a P-gp substrate.

The rapid and strong activation of the cGAS/STING pathway is reported to induce cytotoxic acute responses after DNA damage, but its chronic and constitutive activation offers a pro-survival space for chromosomally instable cancer cells, such as the acquired resistant breast cancer cells. This led us to explore whether STING is involved in paclitaxel resistance. We found that STING was highly expressed in the paclitaxel-resistant breast cancer cells (Figure 6A). The mRNA level of *cGAS*, the key component of the STING pathway, as well as the STING target gene *IFN-β* and its downstream genes *IFIT1* and *IFIT3* were also increased in the MCF7/T cells compared to their parental cells (Figure 6B). RNA-Seq analysis also validated the results. Results of GO (Gene Ontology) analysis (Appendix A) showed that the altered genes were enriched in cell innate immunity and inflammatory response, such as complement activation, type I interferon signaling pathway, response to cytokine, defense response to virus, and so on. GSEA results also pointed to the cytosolic DNA-sensing pathway (Appendix A). When *STING* was knocked down using siRNA, paclitaxel presented higher sensitivity in the MCF7/T cells compared to the paclitaxel treatment alone (Figure 6C and Appendix A). Consequently, the level of IFNβ was reduced after *STING* knockdown (Figure 6D).

Next, we investigated whether S-72 exhibits its antitumor activity via the STING pathway. After S-72 treatment of the resistant cells, the phosphorylation level of STING was reduced, indicating that the STING function was repressed (Figure 6E and Appendix A). Consistently, the STING downstream gene *IFNβ* was downregulated by S-72. Additionally, the *IFNβ*-related genes *IFIT1* and *IFIT3* were reduced in the cells after exposure to S-72 (Figure 6F).

### 2.6. S-72-Induced STING Inactivation Led to Chromosomal Instability

Recent studies have revealed that the STING pathway drives the survival of chromosomally instable cancer cells. Thus, we predicted that S-27 would lead to CIN via STING pathway suppression. Our immunofluorescence study (Figure 7A) showed that S-72 could induce the formation of multi-nuclei and micro-nuclei (red arrows) in both the MCF7 and MCF7/T cells, indicating chromosomal instability. As tubulin polymerization inhibitors, both S-72 and colchicine disseminated and depolymerized microtubule arrangement in both MCF7 and MCF7/T cells, while paclitaxel treatment condensed the microtubules and induced multipolar spindles.

The spindle assembly checkpoints (SACs) BUB1 and BUBR1 are positively correlated with CIN. The results of quantitative real-time PCR showed that S-72 and colchicine enhanced the mRNA levels of *BUB1* and *BUBR1* in both the paclitaxel-sensitive and -resistant cells, but paclitaxel only enhanced the BUB1 and BUBR1 levels in the paclitaxel-sensitive cells (Figure 7B). Furthermore, after S-72 treatment, the phosphorylated H2A.X at Ser139 (γH2A.X), a sign of DNA damage, increased, while p-BRCA1 (Ser1524), the DNA damage repair protein, decreased (Figure 7C).

NF-κB and p21 participate in STING-mediated CIN [27,28]. We checked for these players in the MCF7/T cells after exposure to S-72. S-72, while maintaining CIN, could decrease the expression of p21, which is a positive regulator. The NF-κB pathway was also repressed after S-72 treatment (Figure 7D).

Taken together, we can determine that S-72 led to CIN via blocking of the STING pathway in paclitaxel-resistant breast cancers.

## 3. Discussion

Microtubules possess highly dynamic instability and play crucial roles in several cellular processes, including intracellular trafficking, cell division and cell migration, making them an attractive target for cancer treatment [29]. Among the MTAs, paclitaxel remains the frontline agent for all subtypes of breast cancer treatment, even with the introduction of targeted therapy and immune checkpoint inhibitors [7]. However, the acquired multidrug resistance (MDR) to paclitaxel after long-term use dampens its therapeutic efficacy in clinical practice. To overcome paclitaxel resistance, multiple approaches are currently being developed. The optimization of paclitaxel’s structure at specific positions or modification of the formulations to both improve drug solubility and bioavailability and reduce affinity to drug efflux pumps are popular areas of research. To date, the paclitaxel analogue docetaxel and albumin-bound paclitaxel (nab-paclitaxel) have been applied in the clinic, but drug resistance events still arise after long-term use [30]. The development of mitosis-related kinase inhibitors, such as APC/C^Cdc20^ inhibitors, to overcome mitotic slippage through the inhibition of cell mitosis and to thereby buy more time to induce cell death is another novel therapeutic strategy [31,32]. However, the results of clinical trials show that these drugs commonly cause severe side effects, and their clinical efficacy is poor [7]. Targeting of the metabolic reprogramming induced by taxanes is another way to avoid chemoresistance. Metabolic modulators that target either glycolytic or glutaminolytic enzymes, glucose transporters or lipid and mitochondria metabolism have been shown to sensitize resistant cancer cells to paclitaxel [33]. However, currently, this research is still in the preclinical phase. CBSIs are considered to be the most promising next-generation MTAs, with several significant advantages such as their lesser susceptibility to ABC efflux transporters, higher water solubility and lower molecular weight, all of which contribute to their enhanced oral bioavailability [14]. In addition, the efficacy of several CBSIs is not affected by the expression pattern of β-tubulin, and they are not effluxed by P-gp, rendering them effective in circumventing and combat drug resistance [34]. Several CBSIs, including BNC105P, ABT-751, and VERU-111, have been or are being tested in clinical trials, indicating the importance of CBSIs for effective cancer chemotherapy. Since clinical trials for these CBSIs are still ongoing and inconclusive, we are focused on developing more novel CBSIs that are more effective or that have improved pharmacokinetic properties.

Most of the recently reported novel CBSIs are VERU-111 analogues with common known parent nucleus structures, such as SB226 and CH-2-77 [16,25]. The greatest weakness of VERU-111 and its analogues is the relatively low metabolic stability. The oral bioavailability of VERU-111 is 21–50%, and its water solubility is relatively low [14]. Similarly, SB226 and CH-2-77 also display low half-life time and systemic exposure (AUC) but a high clearance rate, making them relatively insufficient in vivo. Thus, they should be administered with frequent dosing to achieve systemic exposure near the MTD, which may increase the possibility of toxicity and adverse reactions. Furthermore, SB226 and CH-2-77 could only be administered by injection, which may also increase toxicity and reduce convenience in clinical practice. BZML is another newly reported CBSI that can irreversibly inhibit P-gp function via decreasing P-gp expression [24]. However, P-gp expressions are not found to correlate with taxane response in breast cancer patients, and BZML has not entered into clinical trial.

In our efforts to discover novel CBSIs, we identified novel tubulin polymerization inhibitor S-series compounds. Compound 40 demonstrated significant antitumor efficacy in various xenograft models. However, its PK properties, such as its metabolic stability and exposure, need to be optimized further. Since the pyridine moiety is usually considered as a potential metabolically labile site, we replaced it with 1-methylpyrazole based on a bioisosteric strategy.Compound S-72 exhibited more potent nanomolar efficacy, as well as a higher metabolic stability (t_1/2_ 4.07 h), AUC and oral bioavailability (74.1%) (Appendix A). Hence, S-72 has several advantages such as prolonged plasma concentration, reduced toxicity, and convenient frequent dosing, making it more patient friendly and inexpensive. Previously, we reported a uniquely structured small molecule, S-40, as an oral CBSI showing antitumor activities [35]. Herein, S-72, a structurally modified analogue of S-40, was generated to prevent P-gp efflux and to further evaluate the anti-resistance activities.

Firstly, we evaluated the cytotoxic effects of S-72 on both parental breast cancer cells and their paclitaxel-resistant sublines. The control drugs were paclitaxel and colchicine, which target the same tubulin site as S-72 and are not approved for cancer treatment due to their severe toxicity. The CCK8 assay revealed that paclitaxel, colchicine and S-72 all showed great cytotoxicity toward the paclitaxel-sensitive MCF7 and MX-1 cells. In contrast, the MCF7/T and MX-1/T cells showed cross-resistance to paclitaxel and colchicine, while S-72 achieved significant impairment of these two paclitaxel-resistant breast cancer cells. Furthermore, we found that at the same concentration, S-72 caused cell death and inhibited cell colony formation in MCF7/T cells to a superior extent compared with paclitaxel and colchicine, and it also dose-dependently controlled cell invasion and migration in both MCF7 and MCF7/T cells, as reflected by the results of the Transwell assay and the wound healing assay.

Secondly, we verified the anticancer activity of S-72 against paclitaxel-sensitive and paclitaxel-resistant breast cancer cells in vivo. The results showed that the tumor growth rate of the MCF7/T-cell-bearing mice was greater than that of the MCF7-cell-bearing mice, indicating greater malignancy after the cancer cells acquired resistance [13,36]. The therapeutic dose of paclitaxel used for the MCF7-cell-bearing mice failed to inhibit tumor growth in these mice, but S-72 significantly inhibited tumor growth in both the MCF7- and MCF7/T-cell-bearing mice, indicating its anti-resistance effects in breast cancer. Although the body weight of the mice treated with the highest dose of S-72 was significantly reduced, no death occurred throughout the whole administration period. Moreover, in the future, the cumulative toxicity of S-72 could be relieved by prolonging the administration interval or by its encapsulation in nanoparticles or use as a warhead in antibody–drug conjugates. Similar results were shown in the case of the MX-1 and MX-1/T tumor xenograft models, in which the tumors grew more aggressively due to the triple-negative breast cancer phenotypes, and S-72 still effectively inhibited both the paclitaxel-sensitive and paclitaxel-resistant breast cancer cells in vivo.

Thirdly, to confirm the interaction of S-72 with microtubules, we evaluated the effects of S-72 on tubulin polymerization and the cell cycle. The results of the in vitro tubulin polymerization assay, as well as the decreased protein levels of acetylated-α-tubulin, which was reported to stabilize microtubules, demonstrated that S-72 inhibited tubulin polymerization, acting as a CBSI. The immunofluorescence study of the microtubule also showed that S-72 disorganized the microtubule assembly in the MCF7 and MCF7/T cells. This disruption of microtubule dynamics always led to G2/M arrest, further causing cell death [37]. Then, the expression levels of key proteins were detected during the process of cell cycle arrest induced by S-72. The phosphorylation and activation of cdc25 cause the dephosphorylation and activation of cdc2 (also named cyclin-dependent kinase 1, CDK1). The activated cdc2/cyclinB1 complex promotes the transition of G2 to M and has a positive feedback effect on further cdc25 activation [38,39,40]. In this study, S-72 treatment caused the activation of cdc2/cyclinB1, and the cells entered the mitosis phase and accomplished nuclear division but failed to undergo cytokinesis so as to complete the whole cell cycle, resulting in mitosis phase arrest and a significant accumulation of polyploid (>4N DNA) cells. The arrested cells further underwent apoptosis, and the upregulation of p-Histone H3 also indicated that apoptosis occurred primarily in the mitosis phase of the cell cycle. STAT3, a typical oncoprotein that promotes chemo-resistant phenotype formation in breast cancers, is activated to a greater extent through tyrosine 705 residue phosphorylation in MCF7/T and MX-1/T cells [41]. As a CBSI, S-72 inhibits STAT3 tyrosine phosphorylation in both paclitaxel-sensitive and resistant breast cancer cells. It has been reported that MTAs specifically inhibit STAT3 tyrosine phosphorylation via the loss of interaction of STAT3 with tubulin [42,43]. However, the effects of S-72 on STAT3 phosphorylation at serine 727 residue are contradictory. The exact role of serine phosphorylation in STAT3 function is still controversial [44]. Some studies suggested that it is required for the maximal transcriptional activity of STAT3 [45], while others reported the negative modulation of STAT3 activation by serine phosphorylation [46]. The amount of pSer-STAT3 is elevated during mitosis, and thus presumably, it plays an important role in the onset of the mitosis phase [47].

Next, the molecular mechanisms behind the S-72-induced anti-resistant effects were investigated. Recent studies have revealed that CIN, as well as the resulting aneuploidy and chromothripsis, may be the root cause of chemotherapy resistance [8,48,49,50,51,52,53]. Moderate CIN can accelerate the acquisition of drug resistance by increasing karyotypic heterogeneity, which can be exploited by cancer cells in order to survive and adapt to chemotherapy [48,49]. However, enhancing the level of CIN above the tolerance threshold may impair adaptability or exacerbate CIN-associated fitness costs, thus leading to CIN cell elimination and, finally, to anti-cancer resistance [8]. Since microtubules undergo rapid polymerization and depolymerization to ensure the movement of chromosomes, which is essential for cell mitosis and proliferation, MTAs, including S-72, induce the formation of multipolar spindles and depolymerize the microtubules, leading to augmented CIN and, ultimately, cell death [7]. This process has been proven, along with the upregulation of the SACs BUBR1 and BUB1, the DNA damage marker γH2A.X, and the inactivation of the DNA damage repair protein BRCA1, whose gene remains as the wild type in MCF7 and MCF7/T cells [54].

Consistent with the dual functions of CIN in cancer resistance and sensitization, STING acts as a double-edged sword during this complicated process. In paclitaxel-sensitive MCF7 cells, MTAs significantly upregulate STING, as well as its downstream cytotoxic acute type-I interferon (IFN-I) responses. Meanwhile, in MCF7/T cells, cGAS-STING is constitutively activated, leading to chronic IFN-I responses with the overexpression of pro-survival IFN-related DNA damage resistance signatures (IRDSs) such as IFIT1 and IFIT3, supporting the survival of chromosomally instable cancers, thus serving as an antidote for paclitaxel-induced CIN [11,55]. S-72 significantly inhibited cGAS-STING-IFNβ-IRDS signals in the paclitaxel-resistant breast cancer cells, removing the survival signals necessary for CIN cells and causing cell death. Further knockdown of STING sensitized the MCF7/T cells to paclitaxel and S-72, accompanied by a decreased IFNβ concentration, reduced expression of IRDS, and repressed CIN cell survival factors, such as p-NF-κB and p21.

However, the safety of S-72 needs to be evaluated further. The dose-cumulative gastrointestinal toxicity of S-72 may be relieved through its preparation S-72 into nanoparticles, extension of the administration interval, or its use as the payload of the antibody–drug conjugate (ADC). Moreover, considering that cGAS-STING-IFNβ signaling is associated with innate immunity and secondary acquired immunity and that its chronic and persistent activation may result in a tumor immunosuppressive microenvironment, further studies could focus on the effects of S-72 on immune cells, such as myeloid-derived suppressor cells and cytotoxic T lymphocytes, using mice with normal immune systems [18,56].

There are several limitations to our study. First, in vivo studies showed that S-72 at high doses reduced mice body weights, although it could be recovered with the extension of treatment time, and no animal mortality was observed during the study period. Further safety evaluation is needed, and S-72 could be used as a payload in an antibody-drug conjugate or for loading drugs into nanoparticles in the future. Second, our results show that cGAS-STING signaling is involved in paclitaxel resistance of breast cancer, and S-72 inhibits STING activation in the paclitaxel-resistant breast cancer cells, but the study was only conducted in MCF7/T cells, and additional models of paclitaxel-resistant breast cancer should be used to verify these results. We also found that STING signaling was over-activated in MDA-MB-231/T cells compared with its non-resistant parental cells, and further studies may be conducted in MDA-MB-231/T cells to confirm the conclusion. Third, the deep mechanisms of S-72-induced STING inactivation are unclear. As a microtubule targeting agent, S-72 interferes with the formation of mitotic spindles and affects cell mitotic cycle, which may indirectly affect cGAS-STING signaling. The specific mechanism of action needs further elucidation.

In conclusion, the present study reported that S-72 is a brand-new structural CBSI with potent anti-cancer resistant efficacy, relatively good water solubility and high oral bioavailability. S-72 exerts anti-cancer activity against paclitaxel-sensitive and paclitaxel-resistant breast cancers in vitro and in vivo. As a novel CBSI, S-72 inhibits tubulin polymerization, triggers mitosis-phase cell cycle arrest and cell apoptosis and hinders the activation of the JAK2/STAT3 pathway. Moreover, we found a novel clinic-related mechanism for combating breast cancer resistance to paclitaxel, which implies that S-72 restores multipolar spindle formation and causes deadly chromosomal instability in MCF7/T cells by reducing STING signals, and STING suppression further sensitizes MCF7/T cells to paclitaxel and S-72. The excellent anti-cancer activity of S-72 is worthy of further preclinical research as the basis for a promising anti-cancer drug, and the findings of the present study may provide novel ideas and therapeutic strategies to combat drug-resistant breast cancer.

## 4. Materials and Methods

### 4.1. Chemicals and Cell Lines

S-72 was synthesized in our own laboratory (Figure 1). The reagents and solvents were commercially available and used without further purification. As an exception, 4-methyl-3-((4-methylphenyl)sulfonamido)benzoic acid was prepared according to the reported literature [57]. A mixture of 4-methyl-3-((4-methylphenyl)sulfonamido)benzoic acid (0.153 g, 0.5 mmol), (1-methyl-1*H*-pyrazol-4-yl)methanamine (0.073 g, 0.65 mmol), 2-(7-azabenzotriazol-1-yl)-N,N,N′,N′-tetramethyluronium hexafluorophosphate (0.285 g, 0.75 mmol) and triethylamine (0.152 g, 1.5 mmol) in DCM (10 mL) was stirred overnight. Water (50 mL) was added, and the resulting mixture was extracted with DCM (30 mL × 3). The combined organic layers were washed with water (30 mL × 2) and brine (30 mL), dried on anhydrous Na_2_SO_4_, filtered and concentrated. The residue was purified with preparative thin-layer chromatography (silica gel, DCM/methanol = 15:1, *v*/*v*) to afford the product as a white solid (111 mg, 56% yield).

^1^H spectra were recorded using a JOEL 400 MHz NMR spectrometer, and ^13^C spectra were recorded using a Bruker 400 MHz NMR spectrometer. ^1^H NMR (400 MHz, DMSO-*d*_6_) δ 9.62 (s, 1H), 8.73 (t, *J* = 5.7 Hz, 1H), 7.62 (d, *J* = 1.8 Hz, 1H), 7.60–7.56 (m, 2H), 7.52 (d, *J* = 8.3 Hz, 2H), 7.37–7.30 (m, 3H), 7.18 (d, *J* = 8.0 Hz, 1H), 4.24 (d, *J* = 5.7 Hz, 2H), 3.78 (s, 3H), 2.36 (s, 3H), 1.93 (s, 3H). ^13^C NMR (101 MHz, DMSO-*d*_6_) δ 165.1, 143.1, 138.0, 137.7, 137.2, 134.9, 132.9, 130.4, 129.6, 129.5, 126.5, 126.0, 124.8, 119.0, 38.3, 33.4, 21.0, 17.5.

Paclitaxel, colchicine, verapamil and rhodamine 123 (Rh123) were purchased from Topscience (Shanghai, China). For the in vitro studies, the chemicals were prepared in DMSO (Sigma-Aldrich, St. Louis, MO, USA) at a stock concentration of 10 mM, and the final concentration of DMSO in the experiments was less than 0.1% (*v*/*v*).

The human breast cancer cell line MCF7 was purchased from the Cell Resource Center of the Institute of Medical Sciences, Peking Union Medical College. MX-1, and the paclitaxel-resistant sublines MX-1/T and MCF7/T were kept in our lab [26]. The cells were cultured in DMEM supplemented with 10% fetal bovine serum (FBS) and 1% penicillin-streptomycin at 37 °C in a humidified atmosphere with 5% CO_2_. The MCF7/T and MX-1/T cells were cultured in medium supplemented with 50 nM paclitaxel, which we changed to a drug-free medium for at least 1 week before use.

### 4.2. Cell Viability Assay

The cell viability assay was performed to assess the efficacy of S-72 in different breast cancer cell lines. Depending on their growth rate, MCF7, MCF7/T, MX-1, and MX-1/T cells were seeded at a concentration of 1500–2500 cells per well in a 96-well plate. After 24 h, the cells were treated with the test compounds at different concentrations ranging from 0 to 2000 nM in 4–6 replicates for 72 h. After 72 h of incubation, 20 μL of CCK8 reagent was added to each well in dark and incubated at 37 °C for 2 h. Absorbance was recorded at 450 nm. The half maximal inhibitory concentration (IC_50_) values and the resistance index (RI) were calculated to reflect the anti-resistance efficacy of S-72. IC_50_ values were calculated using GraphPad Prism 7 software (San Diego, CA, USA). RI was the ratio of the IC_50_ of the resistant cells to that of the parental cells.

### 4.3. Colony Formation Assay

The colony formation assay was used to assess the cell proliferation ability. When a small number of cells were inoculated into the culture plate, they started to proliferate and become a colony. Thus, the size and number of colonies were determined to evaluate the potential anti-proliferative effects of the compounds. In brief, MCF7 and MCF7/T cells were seeded into six-well plates at a density of 800 cells/well. After overnight attachment, the cells were treated with the test compounds (paclitaxel, colchichine or S-72) at different concentrations (0, 25, 50 and 100 nM) for 24 h. Then, the compounds were removed and shifted to a drug-free medium in order to incubate the cells for another 14 days. The colonies were fixed with 4% formalin solution for 15 min and stained with 0.1% (*w*/*v*) crystal violet for 30 min. The total colony occupied area was calculated by Image J software.

### 4.4. Cell Invasion and Migration Assay

The Transwell assay is a common physical isolation assay that measures the invasion and migration behavior of tumor cells by detecting their permeability. Briefly, MCF7 and MCF7/T cells were added to the top chamber in medium containing 1% FBS and treated with S-72 at concentrations of 0, 12.5, 25 or 50 nM for 18 h. Medium supplemented with 20% FBS was used as the chemoattractant. For the cell invasion assay, medium containing 20% Matrigel was used to pre-coat the chambers. After an 18 h treatment, the penetrated cells on the outer surfaces of the chambers were fixed with 4% formalin solution and then stained with crystal violet and captured using a microscope. The penetrated cell numbers were counted by Image J software.

The wound healing assay is conducted to verify the results of a Transwell assay. A single layer of cells cultured in vitro are scratched, and the effect of the compound on the migration ability of the tumor cells is reflected by their scratch healing state. In brief, MCF7 and MCF7/T cells were seeded into twelve-well plates at a density of 1.5 × 10^5^ cells/well and cultured until 90% confluence was obtained. Then, uniform, identical scratches were created. Subsequently, the cells were treated with the test compounds (paclitaxel, colchicine or S-72 at the concentration of 50 nM) for 18 h and incubated for another 30 h. The culture medium containing 1% FBS was used throughout the assay after cell adherence. The wound closure area was calculated by Image J software.

### 4.5. In Vivo Xenograft Model

All animal experiments were approved by the Ethics Committee for Animal Experiments of the Institute of Materia Medica, Chinese Academy of Medical Sciences & Peking Union Medical College, and were conducted in accordance with the Guidelines for Animal Experiments of Peking Union Medical College. Female athymic nude mice (BALB/c-nu, 6–8 weeks old, 18–20 g) were purchased from HFK Bioscience Co. (Beijing, China) and were housed with a 12 h light–dark cycle with free access to food and water.

MCF7 and MCF7/T tumors from the donor mice were cut into small pieces and implanted subcutaneously into the right flank of the mouse. The MX-1 and MX-1/T cells were suspended in PBS and then inoculated into the right flank of the mouse at a final density of 1 × 10^6^ cells/100 μL. The drug treatments were initiated when the tumor volume reached 100–200 mm^3^. The mice were randomly divided into a vehicle control group, paclitaxel group, and groups set to receive different doses of S-72. S-72 and the vehicle were orally administered once per day. Paclitaxel was administered via intraperitoneal injection every 3 days. For the mice bearing MCF7 and MCF7/T tumors, the doses of S-72 were 2.5, 5 and 10 mg/kg, and the dose of paclitaxel was 24 mg/kg. For the mice bearing MX-1 and MX-1/T tumors, the doses of S-72 were 3.75, 7.5 and 15 mg/kg, and that of paclitaxel was 30 mg/kg. The tumor weight and body weight of the mice were measured every 3 days. The tumor volume was measured using a caliper and calculated as L × W^2^ × 0.5, where L and W represent the larger and smaller dimensions of the tumors, respectively. At the end of the experiment, the mice were sacrificed, and the tumor tissues were obtained for histological and immunoblotting assays.

### 4.6. In Vitro Tubulin Polymerization Assay

The in vitro cell-free tubulin polymerization assay was conducted with a BK011P-Tubulin Polymerization Assay Kit (Cytoskeleton, Denver, CO, USA) according to the manufacturer’s instructions. Microtubule polymerization is followed by fluorescence enhancement due to the incorporation of a fluorescent reporter into the microtubules as polymerization occurs. Thus, the fluorescence intensity can reflect the effect of the compound on the polymerization ability of the microtubules. Briefly, the following reaction system is formulated on ice: 205 μL Buffer 1 + 150 μL Tubulin glycerol buffer + 4.4 μL GTP stock (100 mM) + 85 μL Tubulin stock (10 mg/mL). Then, the respective test compounds (paclitaxel, colchicine or S-72) or DMSO were added to 1/2 area 96-well plate and preheated at 37 °C for 1 min. After quickly adding the formulated mixture, the reaction was immediately initiated at 37 °C using a multifunctional microplate reader (BioTek Synergy H1, Winooski, VT, USA). Changes in the fluorescence intensity (ex = 360 nm, em = 420 nm) were measured via kinetic reading at 37 °C for a total of 2 h.

### 4.7. Flow Cytometry

In order to determine the effects of S-72 on the cell cycle distributions, MCF7 and MCF7/T cells were seeded in 6-well plates and treated with paclitaxel, colchicine and S-72 at a concentration of 100 nM for 24 h and then harvested with trypsin and fixed in ice-cold 70% ethanol overnight. Then, the cells were stained with PI for 30 min. Regarding cell apoptosis analysis, in the early stage of cell apoptosis, the phosphatidylserine in the cell membrane overturns, and Annexin V has a high affinity for it; thus, Annexin V is used to detect the early apoptosis of cells. As a nucleic acid dye, PI cannot penetrate the cell membrane entirely, but for late apoptotic cells and necrotic cells, PI can stain the nucleus through the cell membrane. Therefore, Annexin V and PI can be used in combination to detect cells in different stages of apoptosis. In brief, MCF7 and MCF7/T cells were treated with paclitaxel, colchicine and S-72 at a concentration of 100 nM for 24 h and then doubly stained with Annexin V-FITC and PI. For the intracellular Rh123 uptake assay, MCF7 and MCF7/T cells were treated with 10 μM Rh123 after incubation with the test compounds for 24 h. The analyses were performed using a BD FACS Calibur flow cytometer (BD Biosciences, San Jose, CA, USA). Data analysis was conducted using FlowJo v10 software.

### 4.8. Western Blot

The cells were seeded in 6-well plates and then treated with the test compounds for 24 h. After treatment, cells were harvested and lysed in RIPA lysis buffer supplemented with a protease inhibitor cocktail and phosphatase inhibitor (Topscience, Shanghai, China). After centrifuging at 12,000× *g* for 20 min, the supernatant was collected. The protein concentrations were measured using a BCA Protein Assay Kit (Lablead Biotech, Beijing, China). Approximately 25 μg of protein was resolved via SDS–PAGE and transferred to polyvinylidene difluoride (PVDF) membranes. After blocking for 1 h in 5% BSA in TBST solution, the membranes were incubated overnight at 4 °C with the corresponding primary antibodies: rabbit anti-ac-α-tubulin (K40), anti-α-tubulin, anti-p-cdc25c (T48), anti-cdc25c, anti-p-cdc2 (Y15), anti-cdc2, anti-cyclin B1, anti-p-histone H3(S10), anti-histone H3, anti-cleaved caspase-3, anti-caspase-3, anti-bcl-2, anti-cleaved PARP, anti-PARP, anti-p-STAT3 (Y705), anti-p-STAT3 (S727), anti-p-Histone H2A.X (γH2A.X) (S139), anti-Histone H2A.X, anti-p-BRCA1 (S1524), anti-p21, anti-p-STING (S366), anti-p-STING (S365), anti-STING, anti-p-NF-κB p65 (S536), anti-NF-κB p65, anti-p-TBK1 (S172), anti-P-gp, anti-βIII-tubulin, anti-TAU, anti-C-MYC, anti-p-JAK2 (Y1007/1008), anti-JAK2, and mouse anti-STAT3, anti-TBK1 and anti-β-actin, obtained from Cell Signaling Technology (Danvers, MA). Then, the membranes were incubated with peroxidase-conjugated anti-rabbit or mouse IgG secondary antibody (ZSGB-Bio, Beijing, China). Finally, the protein bands were visualized with the chemiluminescence method (Pierce™ ECL Western blotting Substrate, Thermo Fisher Scientific, Waltham, MA, USA).

### 4.9. Quantitative Real-Time PCR

MCF7 and MCF7/T cells were seeded in 12-well plates and then treated with the test compounds for 24 h. Total RNA was extracted with an RNA-quick purification kit (ES Science, Shanghai, China) according to the manufacturer’s instructions before being converted to cDNA with First-Strand cDNA Synthesis SuperMix (Yeasen, Shanghai, China). Then, the cDNAs were analyzed with a Bio-Rad T100 Thermal Cycler (Bio-Rad, Hercules, CA, USA) using a HieffTM qPCR SYBR Green Master Mix (Yeasen, Shanghai, China). The primers were as follows: BUBR1 (F) 5′-AAATGACCCTCTGGATGTTTGG-3′ and (R) 5′-GCATAAACGCCCTAATTTAAGCC-3′; BUB1 (F) 5′-ACAATCAACGGAGAAAGCATGA-3′ and (R) 5′-CTCCACCACCTGATGCAACT-3′; cGAS (F) 5′-AAGGCCTGCGCATTCAAAAC-3′ and (R) 5′-GAGAAGGATAGCCGCCATGT-3′; IFIT1 (F) 5′-TCTCAGAGGAGCCTGGCTAA-3′ and (R) 5′-CCAGACTATCCTTGACCTGATGA-3′; IFIT3 (F) 5′-CAGAACTGCAGGGAAACAGC-3′ and (R) 5′-TGAATAAGTTCCAGGTGAAATGGC-3′; and GAPDH (F) 5′-GTGGACCTGACCTGCCGTCT-3′ and (R) 5′-GGAGGAGTGGGTGTCGCTGT-3′.

### 4.10. Immunofluorescence

The morphology of the microtubule network and nucleus was determined via an immunofluorescence study. MCF7 and MCF7/T cells were seeded into glass-covered chambers and then treated with paclitaxel, colchicine and S-72 at the concentration of 50 nM for 24 h. After washing with PBS, cells were fixed with 4% paraformaldehyde and permeabilized with 0.1% Triton X-100 in PBS solution. The microtubules were stained with an anti-α-tubulin antibody (Cell Signaling Technology, Danvers, MA, USA) overnight at 4 °C, followed by incubation with Alexa Fluor 488 goat anti-rabbit IgG (Life Technologies, Carlsbad, CA, USA) at room temperature for 1 h. Then, the cells were stained with Hoechst 33342. Cell images were obtained using a Leica TCS SP8 X confocal microscope (Leica, Wetzlar, Germany).

### 4.11. Plasmid Construction and Transfection

To study the role of STING in the paclitaxel resistance of breast cancer, MCF7/T cells were transiently transfected with a mixture of lipofectamine 3000 (Invitrogen, Waltham, MA, USA) and STING-siRNA or negative control siRNA (KeyGEN BioTECH, Nanjing, China) in Opti-MEM (Gibco, ThermoFisher Scientific, Waltham, MA, USA), according to the manufacturer’s instructions. The cells were further treated as designed after their transfection. The sequence information for STING-siRNA was as follows: STING siRNA-1 (F) 5′-GCCUCAUUGCCUACCAGGATT-3′ and (R) 5′-UCCUGGUAGGCAAUGAGGCTT-3′; STING siRNA-2 (F) 5′-CCGGAUUCGAACUUACAAUTT-3′ and (R) 5′-AUUGUAAGUUCGAAUCCGGTT-3′; and STING siRNA-3 (F) 5′-GGCUUUAGCCGGGAGGAUATT-3′ and (R) 5′-UAUCCUCCCGGCUAAAGCCTT-3′.

### 4.12. Sandwich Enzyme Linked Immuno-Sorbent Assay (ELISA)

A DuoSet^®^ ELISA kit (R&D systems, USA) was used to determine the release of IFNβ in the cell culture supernatants, according to the manufacturer’s instructions. Briefly, MCF7/T cells were transfected with *STING*-siRNA and then treated with 100 nM paclitaxel for 24 h. After treatment, the cell culture supernatants were collected and added to the capture-antibody-coated 96-well microplate. After incubating at room temperature for 2 h, the detection antibody was added and incubated at room temperature for 2 h. Then, the streptavidin-HRP and its substrate was used to display color. A multifunctional microplate reader (BioTek Synergy H1, Winooski, VT, USA) was used to read the absorbance at 450 nm.

### 4.13. Statistical Analysis

Data are expressed as the mean ± SD and were analyzed via one-way or two-way ANOVA followed by Dunnett’s multiple comparison test using GraphPad Prism 7 software (San Diego, CA, USA). *p* values < 0.05 were considered to be statistically significant.

## Data Availability

Data is contained within the article and Appendix A.

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
