# Peer review of "S-72, a Novel Orally Available Tubulin Inhibitor, Overcomes Paclitaxel Resistance via Inactivation of the STING Pathway in Breast Cancer"

_pharmaceuticals, 2023, doi:10.3390/ph16050749_

Round 1

Reviewer 1 Report

This study has reported the evaluation of anticancer activity of S-72, a tubulin inhibitor. This inhibitor suppressed the proliferation, invasion and migration of paclitaxel-resistant breast cancer cells via in vitro assays and displayed antitumor activities against the xenografts via in vivo assays.

The figures were illustrated clearly and supported the text body. It is informative to readers and in accordance with aim and scope of Pharmaceuticals.

I suggest accepting this manuscript after a minor revision.

Comments:

1. Section 4.1: the authors provide the NMR data of S-72, however the NMR spectra of this compound are missing in Supplemental material. Please provide.

2. Reference 10 and others: Do not capitalize the first letter of other words except the first word and some other special requirements.

Reviewer 2 Report

The authors worked in this study on the development of resistance through a tubulin inhibitor, given that microtubule-targeting agents are frequently used as active anticancer drugs, and drug resistance always develops with long-term use, especially for compounds such as paclitaxel.

The study is very useful for treating paclitaxel-resistant breast cancer, including a potential strategy to establish sensitivity to paclitaxel.

This is important and successful research.

From my point of view, this study can be accepted after a very minor spell check and the rules of the journal.

Author Response

Dear reviewer,

Thank you for your interest in our study and for giving us the opportunity to revise our manuscript. To further improve the quality of the paper, we have invited a professional medical English editor to totally polish up our manuscript, and we also carefully proof-read the data presentation as well as checked the whole manuscript.

If you have any other questions, please feel free to contact us, and we look forward to hearing from you soon.

Thank you again and best regards!

Reviewer 3 Report

This work introduces a new orally available tubulin inhibitor, S-72, to address the issue of combating paclitaxel resistance in breast cancer. However, the paper has certain shortcomings that need to be addressed to enhance its quality. The following suggestions are offered to improve the paper:

(1) The paper should provide a formal definition of the problem, including an analysis of its challenging features.

(2) A comprehensive overview of the existing research under the same or similar research topics should be presented. This should involve a comparison of their strengths and weaknesses, and the essential differences between this work and the existing ones should be highlighted to support the paper's contributions.

(3) The methods section should clarify the motivations behind the approaches or strategies employed. The paper should offer theoretical analyses, and the problem-specific considerations should be illustrated in detail.

(4) The authors should compare this work with other state-of-the-art approaches to demonstrate its superiority.

(5) The paper should present future work based on thorough discussions of the proposed method's drawbacks.

Reviewer 4 Report

In this paper, Hou Z.Y. et al. report a novel tubulin inhibitor, S-72, which is effective in overcoming paclitaxel resistance through the inactivation of the STING pathway in breast cancer cell lines and mouse xenografts. Resistance development to paclitaxel is a bottleneck in the clinic, and devising effective ways to combat such resistance is crucial to facilitate effective breast cancer treatment. S-72, reported here, may develop into an effective treatment adjuvant to overcome such resistance.

The study is carefully designed, and findings are supported with adequate experimental data. The article is also written well and can be recommended for publication in pharmaceuticals.

Author Response

Dear reviewer,

Thank you for your interest in our study. To further improve the quality of the paper, we have invited a professional medical English editor to totally polish up our manuscript, and we also carefully proof-read the data presentation as well as checked the whole manuscript.

If you have any other questions, please feel free to contact us.

Thank you again and best regards!

Reviewer 5 Report

In this paper, the authors report a novel orally available tubulin inhibitor S-72 that inhibited the proliferation, invasion and migration of paclitaxel-resistant breast cancer cells in vitro and also displayed desirable antitumor activities against the xenografts in vivo. Mechanism study indicated that S-72 typically inhibited tubulin polymerization, triggering mitosis phase cell cycle arrest and cell apoptosis, and suppressing STAT3 signaling. Further studies showed that STING signaling was involved in paclitaxel resistance, andS-72 blocked STING activation in paclitaxel-resistant breast cancer cells. The results presented in this manuscript are of considerable interest. However, the rationale for design of the compound “S-72” is not described in the manuscript. Moreover, the layout of the pictures in the manuscript is a bit messy and should be re-organized properly, for example Figure 3. In addition, as a potential orally available compound, the pharmacokinetic data of S-72 should be supplemented.

There are some typos and grammar errors in the manuscript, some of them are shown below.

1) On page 3, in Figure 1B, the “(nM, 72h ” should be “(nM, 72h)”.

2) There are many “MCF7 and MCF-7” in the context, please unify them. 

3) On page 4, the concentration of drug treatment is not indicated in the footnote of Figure 2B.

4) On page 12, in Figure 7A, treatment concentration of S-72 is not indicated. 

5) On page 15, some phrases in the text are more colloquial, please revise them. Such as “S-72 was synthesized in house.”.

6) On page 15, there are some formatting errors, e.g. in “4-methyl-N-((1-methyl-1H-pyrazol-4-yl)methyl)-3”, H should be italic; in “13C NMR”, the number should be superscribe.

Above all, this manuscript represents an interesting study, which may provide alliterative approach to combat drug-resistant breast cancer. It is recommended a major revision before acceptance.

Round 2

Reviewer 3 Report

The authors present a study on a novel tubulin inhibitor, S-72, and its potential in combating paclitaxel resistance in breast cancer. The paper provides interesting insights into the mechanism of action of S-72 and its preclinical efficacy. However, there are a few areas that require improvement and clarification:

Please provide a more detailed description of the methods and experimental design. This will enable readers to better understand the study and ensure reproducibility. Specifically, information on cell lines, dosages, and experimental conditions should be included.

In the results section, it would be helpful to provide more quantitative data and statistical analyses to support the conclusions drawn. This could include presenting data in tables or graphs, as well as including p-values, confidence intervals, or effect sizes.

The authors mention that S-72 suppresses STAT3 signaling but do not provide sufficient details on how this was determined. Please elaborate on the experiments conducted to assess the effect of S-72 on STAT3 signaling and the results obtained.

The role of STING signaling in paclitaxel resistance is an interesting finding. However, more information on the experimental approach used to investigate this relationship is necessary. Please provide details on the methods used to study the involvement of STING signaling in paclitaxel resistance and how S-72 affects it.

The paper would benefit from a more thorough discussion of the implications of the findings, particularly in the context of existing literature on paclitaxel resistance and other tubulin inhibitors. This will help readers understand the significance of the study and how it contributes to the current understanding of breast cancer treatment.

In the conclusion section, it is essential to discuss any limitations of the study and potential future research directions. This can help guide other researchers who may be interested in building upon the findings presented in this paper.

Please ensure that the language and grammar are carefully reviewed and edited throughout the manuscript for clarity and readability. For example, there are instances where the text switches between past and present tense. Consistent use of tense will improve the overall flow of the paper.

Reviewer 5 Report

I found that the authors have addressed the all criticisms raised by the reviewers. As such, I think the manuscript has been appropriately modified and can be accepted for publication in Pharmaceuticals.

Author Response

Thank you for your reply.

Round 3

Reviewer 3 Report

The authors have addressed all my concerns, and this paper looks good now.